# Experience of personnel involved in dead body management at an apex institute in the aftermath of Odisha triple train collision

**Praisy Joy**[1], **Prajna Paramita Giri**[2], **Madhumita Patnaik**[1]*, **Prabhas Ranjan Tripathy**[1], **Manisha Rajanand Gaikwad**[1], **Pravash Ranjan Mishra**[1], **Sanjukta Sahoo**[1], **Sipra Rout**[1], **Kumbha Gopi**[2], **Dilip Kumar Parida**[3], **Ashutosh Biswas**[3]

1 Department of Anatomy, AIIMS Bhubaneswar, Bhubaneswar, Odisha, India, 2 Department of Community Medicine and Family Medicine, AIIMS Bhubaneswar, Bhubaneswar, Odisha, India, 3 AIIMS Bhubaneswar, Bhubaneswar, Odisha, India

* anat_madhumita@aiimsbhubaneswar.edu.in

## Abstract

### Background

Railway disasters cause huge loss of life and resources. A triple train collision occurred at 7 PM on 2nd June 2023 at Bahanaga, Balasore, Odisha. It was the third deadliest train accident in India with 288 deaths and more than 900 injured. This study aimed to bring out the experience and capture the emotions of the personnel involved in body management of this major accident.

### Materials and methods

This qualitative questionnaire-based study was done between 13/07/2023 and 29/01/2024, involving 47 personnel (including faculty, residents, staff, and students) who managed the deceased bodies. Six open-ended questions dealing with the experiences of the body management team were analyzed using thematic analysis framework method. Quirkos software was used to generate themes and subthemes. Another six questions about self-reported satisfaction levels of the personnel were graded on a Likert scale of 1 to 5.

### Results

Qualitative analysis identified six themes and twelve subthemes. It highlighted critical aspects such as lack of training in managing mass tragedies, inadequate flow of essential supplies, emergency preparedness, and defined standard operating procedures (SOP). Excellent teamwork (91.4% of participants expressed high satisfaction with the teamwork) and on-the-spot decision-making were heralded as strengths. The Likert scale showed that 87.2% of participants rated the overall dead body management at 4 or 5. Furthermore, 93.6% (44/47) of participants gave a score 4 or 5 for Tagging and Embalming.

**Data Availability Statement:** All relevant data are within the manuscript and its Supporting Information files.

**Funding:** The author(s) received no specific funding for this work.

**Competing interests:** The authors have declared that no competing interests exist.

## Conclusions

Teamwork and proper embalming were identified as the top-rated contributors towards effective body management. Recommendations suggested were mock drills and refresher courses in body management for all stakeholders and psychological support to handle the emotional toll of managing mass tragedy. The identification and embalming of dead bodies are an essential humanitarian service and it helped bereaved families to say a final farewell to their loved ones.

## Introduction

A railway accident is defined as an "unwanted or unintended sudden event or a specific chain of such events (occurring during train operation) in which at least one moving rail vehicle is involved, and which has harmful consequences" [1]. On the evening of 2nd June 2023, there was a tragic triple train collision at Bahanaga, district Balasore, Odisha, between the Coromandel Express (Train Number 12841) and a goods train followed by the derailment of 21 coaches of Coromandel Express. The 3 derailed coaches of the Coromandel Express collided with the Bengaluru- Howrah Superfast Express (Train Number 12864) [2]. This led to 288 deaths and over 900 injured victims [3]. The Odisha train accident in 2023 was the third deadliest train accident in India after the 1981 Bihar train derailment (750+ casualties) and the 1995 Firozabad rail disaster (358 casualties) [4].

According to East Coast Railway of India, train accidents are classified as minor (less or equal to 50 casualties inclusive of death and injury, Code Yellow), medium (51 to 99 casualties, Code Orange), and major (more than 100 casualties, Code Red) [5]. Hence Odisha triple train collision accident was thus a major train accident. After 28 years, the nation was facing a massive train accident. All India Institute of Medical Sciences (AIIMS), Bhubaneswar, a teaching medical college granted with the status of 'Institute of National Importance' was given the responsibility of managing the deceased of this train accident.

Any disaster requires management of the injured, management of the deceased and psychosocial support to the affected families. Proper identification and preservation of the deceased bodies is important following a mass tragedy in order to give closure to the family and to tackle the medicolegal aspects. This paper aims to focus on the post-accident proceedings and the lessons learnt to tackle the challenges to better manage any such mass disaster in future. Pooja Chakraborty commented that saving a life is as important as giving dignity to the dead [6]. Hence dead body management is an integral part of disaster response. This is the first study that focusses on the attitude, perception, and opinions of the personnel who were involved in dead body management of this accident.

The research question of this study was to record and analyse the experiences of the personnel involved in the handling and managing the deceased bodies following the triple train collision.

## Materials and methods

### Research approach and design

The present study was a qualitative questionnaire-based study conducted between 13/07/2023 and 29/01/2024 in the Department of Anatomy of a tertiary medical institute in the aftermath

of the Odisha train accident after obtaining clearance from Institutional Ethics Committee (T/IM-NF/Anatomy/23/43).

## Subjects and sampling criteria

The 47 participants included the faculty, the residents and the staff of the Department of Anatomy as well as the undergraduate students of this institute who had managed the deceased bodies of Odisha train accident and were willing to take part in the study.

## Procedure

A google form was created with questions regarding their work experience in managing the deceased bodies of this accident and shared to the email of 43 participants. Each participant was allowed to fill the form only once. We interviewed four participants (the Dissection Hall staff) in the local language (Odia) who were not conversant in English. Their answers were translated by the authors into English and recorded. Filling up the google form or the interview took around 15 to 20 minutes of the participants' time. All the responses were coded and themes and subthemes were generated (Table 2). Data confidentiality and participant anonymity were ensured. This questionnaire was validated by some of the Faculty of the Department of Anatomy for clarity and reliability.

## Method of data collection

The Google form consisted of six open-ended questions regarding their work experience in body management which encouraged the participants to write down in detail about their experiences, feelings and opinion regarding the body management of this accident. The participants had given evocative responses about their personal experience in managing the deceased bodies of this accident. They had given their opinions about the personal and professional lessons learnt from managing this unprecedented tragedy as well as the strengths and weaknesses of the team managing this calamity. Having lived the experience firsthand, the suggestions given by the participants to improve the body management in any future mass calamity situation were specific and practical.

There were an additional six questions about the self-reported satisfaction levels of participants in managing this tragedy which was scored on a Likert scale of 1 to 5 (1 being not managed properly, 2 was just managed, 3 was managed but could do better, 4 was well managed and 5 being very well managed).

## Data management and analysis

The responses of the participants were recorded and analyzed using thematic analysis Framework method. The Quirkos qualitative research software was used to analyze the qualitative responses and to identify patterns to generate themes and subthemes.

## Body management at the tertiary medical institute

The workflow for body management was laid down at the high-level meeting in the institute following the accident chaired by the Executive Director, AIIMS Bhubaneswar (Fig 1). All the stakeholders including Medical Superintendent of AIIMS Bhubaneswar, representatives of Ministry of Health and Family Welfare, Government of India, the Head of Department, Forensic Medicine and the Head of Department, Anatomy, representatives of the Department of Health, Government of Odisha, Odisha Police, officials of the Indian Railways and the Bhubaneswar Municipal Corporation were present in the meeting. The agenda of the meeting

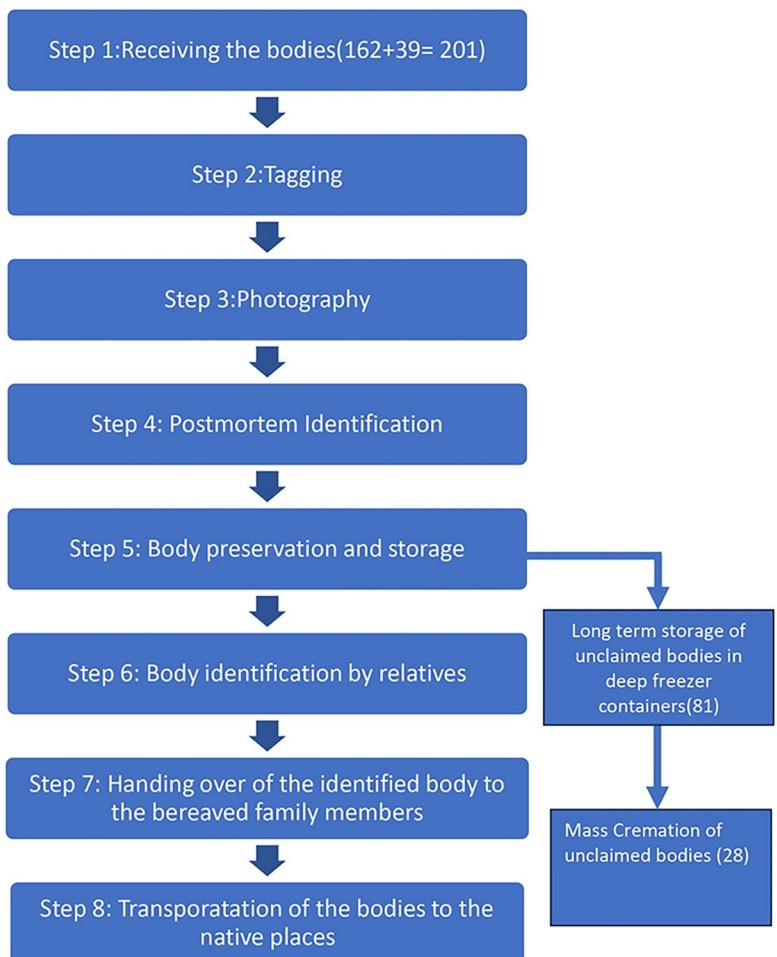

**Fig 1. Workflow of the dead body management during Odisha train accident.** This figure shows the integral steps followed during Odisha train accident 2023.

included timely preservation of the bodies, postmortem examination, sample collection for DNA analysis, body identification, communication with family members and handing over of the bodies to the bereaved family members. In this meeting, it was decided to follow all the necessary protocol pertaining to body preservation, autopsy, sample collection and identification, handing over of the bodies and counselling to the bereaved family members.

The protocol for dead body management was communicated to all involved personnel. The personnel involved were divided into teams and worked on a roster basis with three shifts. At the beginning of each shift, the faculty in charge briefed the personnel. On the early morning of 4th June 2023, 162 deceased bodies were received by the Department of Anatomy. Some bodies were immediately put in mortuary coolers and others kept on blocks of ice to delay decomposition as there were limited number of mortuary coolers available on short notice. Each deceased body was given a unique identification number (UIN) for identification and official documentation. The digital photographs of the deceased bodies were taken by a professional photographer (whole face, frontal view) with the UIN visible in each photograph. Two identical photo albums featuring these photographs were displayed at the help desks. The bereaved families perused the albums for preliminary identification of the deceased. The photographs were continuously displayed on large screen monitors in the foyer of the institute,

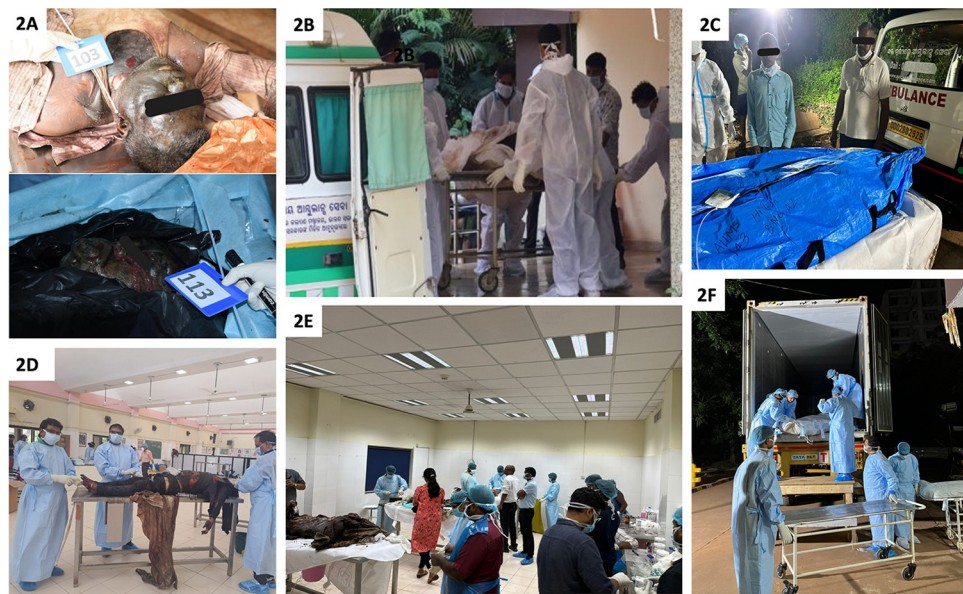

**Fig 2. Representative images of the integral steps followed during body management during Odisha train accident.** 2A) Image showing the Unique Identification Number tagged to the deceased bodies, 2B) Image showing receiving the deceased bodies at the Department of Anatomy, 2C) Image showing handing over of the bodies to the family members, 2D) Image showing embalming of the deceased bodies, 2E) Image showing postmortem examination of the deceased bodies, 2F) Photograph showing transferring the unidentified bodies to the deep freezer container for long term storage.

uploaded to the institute website, and flashed in media to facilitate access to the public. An expert team consisting of Forensic specialists did the post-mortem examination. Personal effects and biological samples like two molar teeth, hair, a piece of sternum and a piece of muscle of the deceased were collected, labeled, and stored individually for DNA analysis. The bodies were embalmed by surface injection and cavity method and then packed in body bags bearing the UIN and stored in deep freezer containers.

Thirty–nine bodies that had been sent to other nearby private medical institutes for postmortem examination and embalming were returned to AIIMS Bhubaneswar for storage and handing over. Once the identity of the deceased was confirmed, the body was placed in a coffin and handed over to Odisha Police after proper documentation. Then the Odisha Police released the body to their family. The deceased bodies were individually transported to their native places by ambulances arranged by the Bhubaneswar Municipal Corporation (BMC). The DNA samples of the unidentified bodies were sent for DNA profiling. These unidentified bodies were kept in deep freezer containers in the premises of the institute till DNA matching could be completed to establish the kinship. On 10<sup>th</sup> Oct 2023, the institute handed over the remaining 28 unidentified dead bodies to the Health Officer of BMC for mass cremation (Fig 2). The information regarding the responses of the participants of this study has been attached as supplementary information (S1 Table).

## Results

The participants of this study had a mean age of 26 ± 9.84 years. The majority of the participants were males (72.3%) and undergraduate students constituted 59.5% of the participants. The sociodemographic details of participants have been compiled in **Table 1**.

**Table 1. Sociodemographic details of participants n = 47.**

| Variables | Categories | Number | Frequency (%) |
|---|---|---|---|
| Age (years) | 18–25 | 29 | 61.7 |
| | 26–40 | 12 | 25.5 |
| | 41–56 | 6 | 12.7 |
| Designation | Undergraduate Medical Students | 27 | 57.4 |
| | Junior Resident, Department of Anatomy | 5 | 10.6 |
| | Tutor, Department of Anatomy | 4 | 8.5 |
| | Dissection Hall Worker | 4 | 8.5 |
| | Associate Professor, Department of Anatomy | 2 | 4.2 |
| | Professor, Department of Anatomy | 1 | 2.1 |
| | Additional Professor, Department of Anatomy | 1 | 2.1 |
| | Assistant Professor, Department of Anatomy | 1 | 2.1 |
| | Senior Resident, Department of Anatomy | 1 | 2.1 |
| | Undergraduate Nursing Students | 1 | .1 |
| Gender | Male | 34 | 72.3 |
| | Female | 13 | 27.7 |

## Qualitative responses

The first question was about the personal experience of the participants in managing the deceased bodies, and the second question asked was the lessons learnt during managing this mass tragedy. In answering these two questions, many had commented about the "unpredictability of life" (P-1,3, 6), and that it was the "first experience" (P-1,5,9,35, 39), "lifetime experience" (P-10, 38), "different experience" (P-12,16) or "heart wrenching" (P-8,14, 24).

Some of the participants have also noted that this was hands-on training for them in managing emergencies. Some participants had said,

"Firstly, I learned not to panic in any situation we face ever in our practice,. . .[sic]" Lastly, by handling such accident, I have gained the confidence to handle situations ever in future in my practice" (P-7).

The participants had expressed "satisfaction", "happiness", "pride" (P-2,3,12) in being part of the team that helped in the management of a "national emergency" (P-1), a "national disaster" (P-10) or a 'terrible accident' (P-13). Others have expressed feeling "sad" (P-2,4) and feeling the pain and anguish of the relatives (P-5,7). One participant had evoked the COVID-19 pandemic experience, "It reminded me of the casualties during COVID pandemic, saw for first time so many bodies together" (P-11).

". . .but preservation was the prime task ahead till there loved ones get to see them for last time" [sic] (P-11).

Regarding the previous experience in managing unprecedented situations, 93.62% of the participants had reported that they had no prior experience. 6.38% of participants (P-1,6 and 8) had answered that they had prior experience. Of which, 2 participants had experienced handling deceased bodies during COVID-19 pandemic and the other participant had handled deceased bodies of a road traffic accident. P-1 had said that breaking bad news to the family was difficult. P-6 had reported that no preservation of deceased bodies was done during her experience with COVID-19. P-8 had said that in her previous experience with a road traffic accident, embalming of the deceased bodies was done by injection method.

## Strengths of the body management team

The ability to work as a "team" was unanimously expressed as the foremost "strength" of the body management team. Further readiness to work round the clock, dedication and sincerity (P-7, 13, 14), taking up responsibility and desire to help the bereaved families (P-12, 13, 14), being "calm" (P-4), leadership qualities of seniors (P-9), the residents and tutors (P-3) have been cited as the institute's strength. Another participant, (P-10) had commented that the "skills in handling cadavers regularly" by the department personnel was one of our strengths. Another respondent (P-8) had observed that "maybe we see death from close compared to others" as a strength in keeping calm and managing the responsibility efficiently.

Others had commented that, "Everyone was ready to do everything. And nobody was complaining and worked for the cause together" (P-5).

"The dedication of all members to work on toes 18–20 hours a day, all of us was always ready for the surprises" (P-7).

## Weakness of the body management team

The participants have listed that lack of fast decision making (P-4) and workplan (P-6,12), absence of work distribution (P-5), lack of communication (P-7, 13,14), scarcity of manpower (P-8,10), inadequate infrastructure to deal with a massive tragedy (P-9,11), lack of prior experience in managing mass calamity situation (P-12) as "weakness" of the of the team.

## Themes and subthemes

From the thematic analysis, six themes and twelve subthemes were generated. The themes and subthemes have been summarised in **Table 2.**

**Theme 1: "Infrastructure".** This theme included two subthemes 1a) "Healthcare workers" and 1 b) "Logistics".

**1a) Under the subtheme "Healthcare workers",** the respondents stressed the need for the mobilization of healthcare personnel from other departments to the Anatomy department to speed up the identification and embalming of the bodies. Some of the participants stated that,

"Mass emergency handling requires shifting of more healthcare personnel from nearby area/ other departments to the actual spot for help" (P-1).

They felt "manpower is essential in managing this kind of situation" (P- 3).

Others have stressed the need for both professionals trained in body management and a group of "support staff" drawn from students, Non-Governmental Organizations (NGO), volunteers to help with tasks requiring no technical expertise. As written by few of them,

"Manpower experts and support staff" (P-11).

According to one of the participants (P-5), "Human resources can be increased. We can use the help of NGO and volunteers for mechanical works. And professional people for others".

**1b) Subtheme "Logistics"** outlined the need for adequate availability of consumables like PPE kit, N95masks, embalming chemicals as well as infrastructure like mortuary cabinets, stretchers for efficient workflow. Some of the suggestions given by the participants were,

"All essential supplies needed for embalming, identifying and storing of deceased bodies should be readily available" (P-14).

"Improve cold storage facility" (P-19)

**2. Theme "Training".** Under this theme, there were three subthemes:

2 a) "psychological",

**Table 2. Themes and subthemes generated from qualitative analysis.**

| Themes | Sub Themes | Codes |
|---|---|---|
| **Infrastructure** | **Healthcare workers** | Lack of manpower |
| | | Shifting of more healthcare personnel from nearby area/ other departments |
| | | Involve people from all departments |
| | | Manpower is essential in managing this kind of situation |
| | **Logistics** | Shortage of stretchers |
| | | More mortuary cabinets to store bodies |
| | | Storage capacity and whether to embalm all bodies |
| | | PPE kits and N95 masks and protective materials in a storage |
| | | All essential supplies needed for embalming, identifying, and storing of deceased bodies |
| **Training** | Psychological | Lack of training in the mental, emotional to manage the deceased bodies |
| | | I learnt some psychological things to handle through this accident |
| | | Emotional moment and was a challenge to me |
| | Body management | Lack of training in body management |
| | Workplan | a short brief about the work plan and further procedures. |
| | | Proper distribution of work |
| **Emergency situation** | Emergency preparedness | Preparedness to manage such situations |
| | Working hours | Day and night tiring duties on a national emergency tragedy |
| | Prevention measures | Proper prevention measures to be taken to avoid this kind of tragedy. |
| **Team Work** | Inter department coordination | working as a team will help to sort out the difficult situations |
| | | Leadership qualities, teamwork, dedication |
| | | Team under unprecedented circumstances, but coordination and communication |
| | | Proper tagging, photography, storing the bodies in sequence and embalming |
| | | NGO and volunteers for mechanical works |
| | | Co-ordination between all the departments, police, and municipal corporation |
| **Decision making** | Leadership | No authoritative decision |
| | Electronic device | Alert and careful during handle the electronic device. |
| | | Network and communication is much needed as this time |
| **SOPs** | Guidelines | Policies regarding management of mass casualties (including both injured and deceased) |

2 b) "body management" and

2 c) "workplan".

**2 a) Subtheme "psychological":** Most participants have reported feeling very emotional about the tragic accident and felt the necessity for psychological support for personnel managing such massive tragedies. Several participants have expressed that,

"Still the pain of relatives was also a dreadful thought" (P-5).

"Managing such stressful situation, performing our duties in chaos, screaming of relatives for the loved ones, has taught me how be polite and humble at the same time" (P-7).

"It was emotional moment and was a challenge to me" (P-9).

"At first I am stunned by seeing the dead bodies. Even I thought life has no meaning. . ."(25)

"Firstly it was disturbing to see such a large number of dead bodies in deformed condition. . ." (P-36)

**2 b) Subtheme "body management"** Some participants emphasized the need for training in body management to handle the such mass calamity situations.

"All staff, students, residents and faculty should be trained in body management" (P-14)

2 c) **Under subtheme "workplan"** Some participants felt that a proper "workplan" and distribution of responsibilities would ensure more efficient management of emergency situations. Few of them have said that,

"...do a division of duties and head in charge for each duty. Human resources will be divided according to duties" (P-5).

"Proper distribution of work" (P-8).

**3. Theme "Emergency situation".**   The three subthemes under this theme were:

3 a) "emergency preparedness",

3 b) "working hours" and

3 c) "prevention measures"

**3 a) Under the subtheme "emergency preparedness",** the respondents have written that disaster prevention as well as emergency preparedness go hand in hand. According to some participants,

"Any department irrespective of clinical or nonclinical should be prepared to manage an emergency situation" (P-12).

"Preparedness in any such situations is my take home message" (P-13).

**3 b) Under the subtheme "working hours"** the participants have commented on hectic working hours during the body management duties but still expressed contentment in fulfilling their duties as a member of the body management team. A few have commented that,

"Day and night tiring duties on a national emergency tragedy...[sic]" (P-1).

"It was a hectic week for the department as we had to receive the deceased, followed by tagging, forensic examination, embalming and dispatching. I also felt proud for doing a part in this disaster management that shook the whole nation" (P-12).

**3 c) Under subtheme "prevention measures".**   One participant has stated that "Prevention is better than cure. Proper prevention measures to be taken to avoid these kind of tragedy" (P-2).

**4. Theme "Team work".**   Had only one subtheme "Inter department coordination".

This theme highlighted that proper intra and interdepartmental "coordination" and "communication" to be essential for efficient workflow and successful management of mass calamities. Many participants articulated this sentiment,

"...working as a team will help to sort out the difficult situations" (P-1)

"In any kind of mass tragedy, each person handling the situation is as important as you, his work is as important as ours, to establish a better network and communication is much needed as this time [sic]" (P-7).

"We should work on to develop better coordination between all the working departments... In these kinds of difficult unexpected hours, we should be holding hands with each other, we should be motivating our sub staffs that formed the strong pillar while working with the decomposed bodies [sic]" (P-7).

"...Mass casualty removes barriers between people of different ranks and places" (P-10).

"But we worked as a team with determination to ensure a dignified final farewell to the deceased and provide some closure to the bereaved family members" (P-14).

**5. Theme "Decision Making".**   It had two subthemes

5 a) "Leadership" and

5 b) "Electronic device".

Participants have asserted the need for "authoritative decision" and proper communication.

Many participants have emphasized that leadership of higher administration and departmental seniors had inspired them. Some participants have opined that,

"The leadership qualities, teamwork, dedication all were phenomenal!!"(P-4)

"During mass tragedy with limited resources, with coordination and under a good leadership it's possible to manage" (P-5).

"Proper communication, good leadership, mutual understanding and cooperation. . ." (P-12).

**6. Theme "SOP".** Under this theme, only one subtheme "guidelines" was there. The participants had pointed out the need for standard operating procedure (SOP) to manage mass tragedies.

"We need better policies regarding management of mass casualties (including both injured and deceased). . ." (P-6)

"Proper SOP should be created and followed" (P-13)

## Likert scale analysis of participant satisfaction

The next six questions were scored on a Likert scale of 1–5 about the satisfaction levels of the participants reflecting the efficiency of various integral steps followed during body management like

1. photography,

2. tagging,

3. embalming,

4. data handling,

5. handing over bodies to the police,

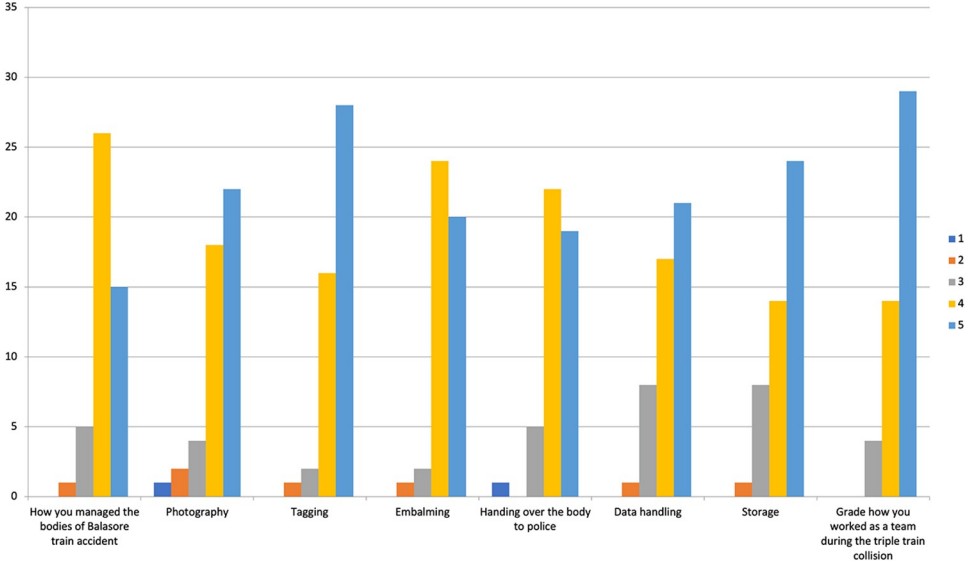

**Fig 3. Graph showing the analysis of Likert scale about satisfaction level of the participants during dead body management.** In the Likert scale of 1 to 5, 1 represents not being managed properly, 2 represents was just managed, 3 represents managed but could do better, 4 represents well managed, 5 represents was very well managed.

6. storage [7] (**Fig 3**).

Overall efficiency of dead body management of this accident was given the highest score of 5 by 31.9% (15) participants and 55.3% (26) gave a score 4.

1. The quality of Photography was given a score of 5 by 46.8% (22) of participants and 38.3% (18) of participants gave a score of 4.

2. Tagging of the dead bodies was graded with the highest score 5 by 59.5% (28) of participants and a score of 4 was given by 34% (16).

3. Embalming of the dead bodies, was given the maximum score of 5 by 42.5% (20) participants and 51% (24) scored it at 4.

4. data handling: 93.6% (44) of the participants were of opinion that data handling was well to very well managed, (score 4–5).

5. handing over of bodies: Eighteen participants (38.3%) felt that handing over of bodies to the police was very well managed and seventeen participants (36.1%) felt that it was well managed.

6. Storage of dead bodies: 51% (24) participants gave a score of 5 for storage of dead bodies and 29.7% (14) participants gave a score of 4 for the same.

The efficiency of the Team during the accident management was scored as highly satisfactory by 61.7% (29) participants, satisfactory by 29.7% (14) participants and just satisfactory by 8.5% (4) participants.

## Discussion

### Railway disaster

In 2005 Indian Railways, defined railway disaster as "a serious train accident or an untoward event of grave nature, either on railway premises or arising out of railway activity in that area, due to natural or manmade causes, that may lead to loss of many lives and/or grievous injuries to a large number of people, and/or severe disruption of traffic etc, necessitating large scale help from other Government, Non-government and Private Organizations" [8].

According to Forsberg et al., the Indian Railway is the world's fourth largest railway and accident are inevitable due to the long rail route [9]. They also mentioned that India has the most rail passenger kilometers travelled per year (838 billion passenger-kilometers) which contributed to fatal railway accidents [9].

### Trends in railway accidents

Globally there has been an increasing trend of railway accidents from 1980's and similar trend has been noted in India. Sarkar and Sarkar reported that in India, from 1951 till 1970, the incidence of train accidents was decreasing and was lowest from 1971 to 1980. From 1981 to 1990, the incidence showed a steep rise. Between 1991 to 2020, the incidence of train accidents had reduced [10]. Better train carriage construction and use of technologically advanced crash avoidance systems are now in place for safety of rail transport [11].

During the last two decades, there has been a steep decline in consequential train accidents from 473 in 2000–01 to 48 in 2022–23 [9]. Though the number of train accidents have decreased in India from 2000–2023, yet the severity of the train accidents has risen with increasing trend of causalities and huge economic loss per accident [12].

## Causes of railway accidents

Human error, equipment failure and sabotage were found to be the major causes of railway accidents. Kaur stated that the infrastructure and carrying capacity constraints also contribute to train accidents in India [13].

## Railway accident prevention and safety systems

Majority of the railway accidents are caused due to human errors. Human errors are mainly caused due to increased workload [14]. Rajabalinejad et al., proposed a "system approach in which both railway personnel and technology work together to reduce major railway accidents [14].

Railway Accident prevention requires use of technology like anti-collision devices and railway electronic signalling system. Evans et al., in his study on European railways reported that "the proportion of accidents caused by signals passed at danger fell from 40% in the 1990s to 21% in the 2010s. This may be due to the increasing deployment of train protection systems" [15]. Eftekhari et al., mentioned that the use of advanced technology, the improvement of work processes and efficient management of human resources are an effective step in reducing the risk of accidents [16].

Indian railways have indigenously developed an automatic train protection system called Kavach to prevent collisions and achieve zero accidents [17]. Kavach uses Artificial Intelligence, the Internet of Things, and sensor-based systems to provide real-time data on the location and movement of trains [18]. This innovative technology can automatically apply the brakes and avoid collisions [19].

The participants of the present study have commented on the need for proper preventive measures to be taken to avoid such mass tragedy under the theme emergency and subtheme preventive measures.

## Themes and subthemes that emerged from this study

In this study, under the theme "infrastructure" and subtheme health care workers, the participant responses were similar to Cordner et al., study which expressed that first responders and experts are required for managing the situation well [20].

In our study, under the theme of Training, the participants felt that training in managing the work and its psychological impact was needed. According to National Disaster Management Guidelines India 2010, the core principles in the management of the dead include preparedness and training, capacity building and storage plus preservation of the deceased [21]. Gupta et al., in 2015 have stated that readiness and training of personnel for handling emergencies is much needed [22]. Chakraborty stated that all the stakeholders should be trained and educated about their role in disaster management [6].

Under the theme of training and sub-theme "psychological" the participants have stressed the need for psychological training to handle the mental and emotional impact of handling mass disaster situations. Brooks et al. did a semi-structured interview followed by a thematic analysis of the findings on forty participants, who were emergency workers (police, firefighters, body handlers, and rescue workers) exposed to the psychological impact of disaster management. They state that most personnel have little to no training in handling the psychological fallout of disaster management. The participants of that study like ours stressed on the need for psychological training and mental health education to cope with the impact of mass disaster situations [23].

Cordner and Ellingham in their work brought out the fact that identification of the bodies played a very important role in management of dead bodies during mass calamities. They also reported that Photographs and DNA analysis played a vital role in body management [20]. In

our study, under the theme SOP, the participants have stressed that DNA profiling for body identification was essential.

Evans et al., reported that effective communication and following standard protocols are the two apples of eye in managing disaster [15]. These findings were observed in our study under the theme Teamwork and SOP respectively.

## Recommendations

1. Training the team: The authors recommend that periodic mock drill and short courses on body management for the personnel to be involved in body management.

2. Psychological support and assessment: The emotional impact of such a tragic event requires proper psychological assessment. Necessary psychological support should be provided to the involved personnel as most of them might not have handled a mass casualty situation before.

## Conclusion

Disaster preparedness and management requires a collaborative approach which involves effective communication among both intra and interdepartmental teams. Every medical institute should have a core committee for disaster management which should include representatives of Department of Anatomy, Forensic Medicine, Emergency Medicine, Community medicine, Psychiatry, Nursing, Lab Medicine and Hospital Administration. Teamwork and effective communication go a long way in managing unprecedented mass tragedies. The identification and embalming of dead bodies was an essential humanitarian service and it helped bereaved families to say a final farewell to their loved ones. This also ensured a dignified farewell to the deceased, which is their basic right.

## Supporting information

**S1 Table. This table contains the detailed responses of the participants to the questions regarding body management.**
(XLSX)

**S1 File.**
(DOCX)

## Acknowledgments

The authors would like to thank the Ministry of Health and Family Welfare, Government of India, State Government of Odisha, Administration, AIIMS Bhubaneswar, Department of Forensic Medicine, AIIMS Bhubaneswar, Bhubaneswar Municipal Corporation, Odisha Police, Indian Railways, Anatomy and Forensic teams from other institutes, various NGOs for their service and helping hands during time of need.

We would like to extend our heartfelt condolences to the families of the deceased of the Odisha train accident.

## Author Contributions

**Conceptualization:** Praisy Joy, Madhumita Patnaik.

**Data curation:** Prajna Paramita Giri, Kumbha Gopi.

**Formal analysis:** Prajna Paramita Giri, Kumbha Gopi.

**Methodology:** Praisy Joy, Madhumita Patnaik, Prabhas Ranjan Tripathy, Manisha Rajanand Gaikwad, Pravash Ranjan Mishra, Sanjukta Sahoo, Sipra Rout, Dilip Kumar Parida, Ashutosh Biswas.

**Software:** Prajna Paramita Giri, Kumbha Gopi.

**Supervision:** Dilip Kumar Parida, Ashutosh Biswas.

**Validation:** Sanjukta Sahoo, Sipra Rout.

**Writing – original draft:** Praisy Joy, Madhumita Patnaik.

**Writing – review & editing:** Prabhas Ranjan Tripathy, Manisha Rajanand Gaikwad, Pravash Ranjan Mishra, Ashutosh Biswas.

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
