## [Decision Letter · Decision Letter 0]

6 Sep 2024

PONE-D-24-31730Experience of personnel involved in dead body management at an apex institute in the aftermath of Odisha triple train collision.PLOS ONE

Dear Dr. Patnaik,

Thank you for submitting your manuscript to PLOS ONE. After careful consideration, we feel that it has merit but does not fully meet PLOS ONE’s publication criteria as it currently stands. Therefore, we invite you to submit a revised version of the manuscript that addresses the points raised during the review process.

We look forward to receiving your revised manuscript.

Kind regards,

Ramesh Athe, PhD

Academic Editor

PLOS ONE

**Journal Requirements:**

2. In the online submission form, you indicated that Data cannot be shared publicly because of confidentiality. Data are available from the corresponding author on request.

Reviewers' comments:

Reviewer's Responses to Questions

**Comments to the Author**

1. Is the manuscript technically sound, and do the data support the conclusions?

Reviewer #1: Yes

Reviewer #2: Yes

2. Has the statistical analysis been performed appropriately and rigorously? 

Reviewer #1: N/A

Reviewer #2: Yes

3. Have the authors made all data underlying the findings in their manuscript fully available?

Reviewer #1: No

Reviewer #2: Yes

4. Is the manuscript presented in an intelligible fashion and written in standard English?

Reviewer #1: Yes

Reviewer #2: Yes

5. Review Comments to the Author

**Reviewer #1: **I would like to congratulate the authors on conducting a thorough qualitative study on the experience of personnel involved in managing a disaster. I have a few comments that are as follows:

I would like to congratulate the authors on conducting a thorough and well-thought qualitative analysis of one of the worst accidents in the country of India.

Following are a few suggestions that will improve the quality of the manuscript:

1. Lines 74 – 77: Include the train number for all the trains involved.

2. Lines 84 – 85: Rephrase to “Odisha triple train collision accident was thus a major train accident”

3. Lines 86 – 87: Rephrase to “All India Institute of Medical Sciences (AIIMS), Bhubaneswar, a teaching medical college granted with the status of ‘Institute of National Importance’ was given the responsibility of managing the deceased of this train accident”.

4. Line 100: Rephrase to “The present study was a qualitative questionnaire-based …”.

5. Line 102: Rephrase to “… accident after obtaining clearance from the Institutional Ethics Committee …”.

6. Line 109: Rephrase to “A google form …”.

7. Line 114: Rephrase to “Participants’ time”.

8. Lines 114-115: The table for the themes and subthemes should be cited here.

9. Line 116: Please elaborate on what the authors mean by validation of the questionnaire by experts.

10. Line 135: Ensure that the subtitle is in sentence case, or capitalize each first alphabet of all the words.

11. Lines 136-137: Please elaborate on the high-level meeting mentioned (the agenda for the meeting, the participants involved, the conclusion of it, etc.). Also, the Figure 1 is of poor quality, and should be resubmitted in higher resolution (>800 DPI).

12. Lines 147-148: Rephrase to “An expert team consisting of …”.

13. Line 148: Mention what biological samples were collected for DNA analysis

14. Line 152: Replace “us” with “AIIMS Bhubaneswar” or “THC where the study was conducted”.

15. Line 163: Figure 2 is of poor quality, and should be resubmitted in higher resolution (>800 DPI).

16. Table 1: Add (years) next to Age; Insert an “em-dash” (–) between all the numbers in the age row; Change the designation of MBBS Students to Undergraduate Medical Students; Write full forms of DH and H in the table’s legend; Change the designation of nursing students to Undergraduate Nursing Students; Add departments of all the consultants and residents.

17. Lines 193 – 194: Rephrase to “COVID-19”.

18. Lines 198 – 199: Authors should quote the responses of the 6.38% of individuals who had prior experience in handling DVM cases.

19. Line 206: Rephrase to “the institute’s strength”.

20. Line 209 – 210: Rephrase to “Another participant had commented …”.

21. Line 332: Figure 3 is of poor quality, and should be resubmitted in higher resolution (>800 DPI). Since the figure is a graph, I suggest inserting it in the word document as a graph itself instead of saving it as an image and then inserting it.

22. Citation for Likert scale should be inserted wherever necessary.

23. Line 362: Rephrase to “Forsberg et al.” instead of “Forsberg R et al.”.

24. Line 368: Rephrase to “Sarkar and Sarkar”.

25. Line 379: Rephrase to “Kaur” instead of “Manpreet Kaur”.

26. Line 383: Rephrase to “Rajabalinejad et al.”.

27. Line 387: Rephrase to “Evans et al.”.

28. Line 398: Rephrase to “The participants of the present study…”.

29. Line 405 – 412: The font type has changed in this paragraph, make it uniform.

30. Line 410: Rephrase to “Chakraborty”.

31. Line 422: Rephrase to “Cordner and Ellingham…”.

32. Line 427: Rephrase to “Evans et al.”.

33. Line 504: The format of the reference is incorrect, and should be revised.

Furthermore, the authors should include their questionnaire in the form of supporting material, or deposit it in a public repository.

**Reviewer #2: **1. The background section of this study provides a rich and detailed description of the accident, but it is recommended that the uniqueness of the study and its potential contribution to the management of similar disasters in the future be further emphasized in the elaboration process.

2. Although the study is based on survey analysis, it is recommended that an attempt be made to analyze some of the results quantitatively, e.g., through statistically specific graphs and charts to enhance the persuasiveness and visualization of the results.

3. How to make sure that all staff strictly follow the regulation process during the corpse disposal process is a concern.

4. In line 138, it is mentioned that “some bodies were immediately placed in a morgue cooler and others were placed on ice."  Would the different ways of handling the bodies have an impact on the subsequent autopsy investigation?

5. In the recommendation section in line 430, the recommendations are enriched to include, for example, physical health checks, vocational rehabilitation, and social support to make the recommendations more practical.

6. Overall, the quality of the writing is good, but it is recommended that the author proofread carefully to eliminate any grammatical or spelling errors.

6. PLOS authors have the option to publish the peer review history of their article (what does this mean?). If published, this will include your full peer review and any attached files.

Reviewer #1: **Yes: **Rutwik Shedge

Reviewer #2: No

---

## [Author Response · Author response to Decision Letter 0]

3 Oct 2024

Response to Reviewers Comments

Sl.no. Reviewer 1’s comments Authors’ response

1. Lines 74 – 77: Include the train number for all the trains involved • Coromandel Express (Train no:12841) 

• Bengaluru- Howrah Superfast Express(Train no:12864)

• Goods train: train number not available

2. Lines 84 – 85: Rephrase to “Odisha triple train collision accident was thus a major train accident” Corrected as per reviewer’s comment

3. Lines 86 – 87: Rephrase to “All India Institute of Medical Sciences (AIIMS), Bhubaneswar, a teaching medical college granted with the status of ‘Institute of National Importance’ was given the responsibility of managing the deceased of this train accident”. Corrected as per reviewer’s comment

4. Line 100: Rephrase to “The present study was a qualitative questionnaire-based …”. Corrected as per reviewer’s comment

5. Line 102: Rephrase to “… accident after obtaining clearance from the Institutional Ethics Committee …”. Corrected as per reviewer’s comment

6. Line 109: Rephrase to “A google form …”. Corrected as per reviewer’s comment

7. Line 114: Rephrase to “Participants’ time”. Corrected as per reviewer’s comment

8. Lines 114-115: The table for the themes and subthemes should be cited here. Corrected as per reviewer’s comment

9. Line 116: Please elaborate on what the authors mean by validation of the questionnaire by experts. The questionnaire was validated by some of the faculty of the Department of Anatomy (PRM, SS, SR) for clarity and reliability.

10. Line 135: Ensure that the subtitle is in sentence case, or capitalize each first alphabet of all the words. Corrected as per reviewer’s comment

11. Lines 136-137: Please elaborate on the high-level meeting mentioned (the agenda for the meeting, the participants involved, the conclusion of it, etc.). Also, the Figure 1 is of poor quality, and should be resubmitted in higher resolution (>800 DPI). Corrected as per reviewer’s comment and Figure 1 has been changed according to reviewer’s comment.

12. Lines 147-148: Rephrase to “An expert team consisting of …”. Corrected as per reviewer’s comment

13. Line 148: Mention what biological samples were collected for DNA analysis Corrected as per reviewer’s comment

14. Line 152: Replace “us” with “AIIMS Bhubaneswar” or “THC where the study was conducted”. Corrected as per reviewer’s comment

15. Line 163: Figure 2 is of poor quality, and should be resubmitted in higher resolution (>800 DPI). Corrected as per reviewer’s comment

16. Table 1: Add (years) next to Age; Insert an “em-dash” (–) between all the numbers in the age row; Change the designation of MBBS Students to Undergraduate Medical Students; Write full forms of DH and H in the table’s legend; Change the designation of nursing students to Undergraduate Nursing Students; Add departments of all the consultants and residents. Corrected as per reviewer’s comment

17. Lines 193 – 194: Rephrase to “COVID-19”. Corrected as per reviewer’s comment

18. Lines 198 – 199: Authors should quote the responses of the 6.38% of individuals who had prior experience in handling DVM cases. Details have been added to the manuscript as per reviewer’s comment

19. Line 206: Rephrase to “the institute’s strength”. Corrected as per reviewer’s comment

20. Line 209 – 210: Rephrase to “Another participant had commented …”. Corrected as per reviewer’s comment

21. Line 332: Figure 3 is of poor quality, and should be resubmitted in higher resolution (>800 DPI). Since the figure is a graph, I suggest inserting it in the word document as a graph itself instead of saving it as an image and then inserting it. Corrected as per reviewer’s comment

22. Citation for Likert scale should be inserted wherever necessary. The citation has been inserted as per reviewer’s comment

23. Line 362: Rephrase to “Forsberg et al.” instead of “Forsberg R et al.”. Corrected as per reviewer’s comment

24. Line 368: Rephrase to “Sarkar and Sarkar”. Corrected as per reviewer’s comment

25. Line 379: Rephrase to “Kaur” instead of “Manpreet Kaur”. Corrected as per reviewer’s comment

26. Line 383: Rephrase to “Rajabalinejad et al.”. Corrected as per reviewer’s comment

27. Line 387: Rephrase to “Evans et al.”. Corrected as per reviewer’s comment

28. Line 398: Rephrase to “The participants of the present study…”. Corrected as per reviewer’s comment

29. Line 405 – 412: The font type has changed in this paragraph, make it uniform. Corrected as per reviewer’s comment

30. Line 410: Rephrase to “Chakraborty”. Corrected as per reviewer’s comment

31. Line 422: Rephrase to “Cordner and Ellingham…”. Corrected as per reviewer’s comment

32. Line 427: Rephrase to “Evans et al.”. Corrected as per reviewer’s comment

33. Line 504: The format of the reference is incorrect, and should be revised. Corrected as per reviewer’s comment

Sl.No. Reviewer 2’s Comment Author’s Response

1. 1. The background section of this study provides a rich and detailed description of the accident, but it is recommended that the uniqueness of the study and its potential contribution to the management of similar disasters in the future be further emphasized in the elaboration process. As per Reviewer’s comment, lines 89 to 92 has been added to the manuscript.

2. Although the study is based on survey analysis, it is recommended that an attempt be made to analyze some of the results quantitatively, e.g., through statistically specific graphs and charts to enhance the persuasiveness and visualization of the results. Figure 3. Graph showing the analysis of Likert Scale about satisfaction level of the participants during dead body management is a graph representing the study results conclusively (quantitatively).

3. How to make sure that all staff strictly follow the regulation process during the corpse disposal process is a concern. The protocol for dead body management was communicated to all involved personnel. The personnel involved were divided into teams and worked on a roster basis with three shifts. At the beginning of each shift, the faculty in charge briefed the personnel. 153-155 has been added to the manuscript. If permitted we will add more photographs of the body management

4. In line 138, it is mentioned that “some bodies were immediately placed in a morgue cooler and others were placed on ice." Would the different ways of handling the bodies have an impact on the subsequent autopsy investigation? Lines 138-140 were removed from the manuscript This study is based on the attitude, perception, and opinions of the personnel who were involved in dead body management of this accident. We have not studied the outcome of autopsy; 

5. In the recommendation section in line 430, the recommendations are enriched to include, for example, physical health checks, vocational rehabilitation, and social support to make the recommendations more practical. The physical health checkup of all employees and students is done mandatorily at time of recruitment /joining. And we have EHS (Employee Health Scheme) which takes care of physical and mental health of all employees .The recommendation about vocational rehabilitation, and social support is beyond the scope of this paper.

6. Overall, the quality of the writing is good, but it is recommended that the author proofread carefully to eliminate any grammatical or spelling errors. We have carefully proof read the manuscript.

---

## [Editor Report · Decision Letter 1]

10 Oct 2024

Experience of personnel involved in dead body management at an apex institute in the aftermath of Odisha triple train collision.

PONE-D-24-31730R1

Dear Dr. Patnaik,

We’re pleased to inform you that your manuscript has been judged scientifically suitable for publication and will be formally accepted for publication once it meets all outstanding technical requirements.

Kind regards,

Ramesh Athe, PhD

Academic Editor

PLOS ONE
---

## [Editor Report · Acceptance letter]

19 Oct 2024

PONE-D-24-31730R1 

PLOS ONE

Dear Dr. Patnaik, 

I'm pleased to inform you that your manuscript has been deemed suitable for publication in PLOS ONE. Congratulations! Your manuscript is now being handed over to our production team.

Kind regards, 

on behalf of

Dr. Ramesh Athe 

Academic Editor

PLOS ONE